# Genome-Wide Profiling of Laron Syndrome Patients Identifies Novel Cancer Protection Pathways

**DOI:** 10.3390/cells8060596

**Published:** 2019-06-15

**Authors:** Haim Werner, Lena Lapkina-Gendler, Laris Achlaug, Karthik Nagaraj, Lina Somri, Danielle Yaron-Saminsky, Metsada Pasmanik-Chor, Rive Sarfstein, Zvi Laron, Shoshana Yakar

**Affiliations:** 1Department of Human Molecular Genetics and Biochemistry, Sackler School of Medicine, Tel Aviv University, Tel Aviv 69978, Israel; lenalapkina@gmail.com (L.L.-G.); laris1010.ab@gmail.com (L.A.); mailkartz@gmail.com (K.N.); lina_somri@hotmail.com (L.S.); danielle.yaron@gmail.com (D.Y.-S.); rives@tauex.tau.ac.il (R.S.); 2Yoran Institute for Human Genome Research, Tel Aviv University, Tel Aviv 69978, Israel; 3Bioinformatics Unit, George Wise Faculty of Life Sciences, Tel Aviv University, Tel Aviv 69978, Israel; metsada@tauex.tau.ac.il; 4Endocrine and Diabetes Research Unit, Schneider Children’s Medical Center, Petah Tikva 49202, Israel; laronz@clalit.org.il; 5David B. Kriser Dental Center, Department of Basic Science and Craniofacial Biology, New York University College of Dentistry, New York, NY 10010-4086, USA; sy1007@nyu.edu

**Keywords:** insulin-like growth factor 1 (IGF1), IGF1 receptor (IGF1R), growth hormone receptor (GH-R), Laron syndrome, cancer protection, thioredoxin-interacting protein (TXNIP)

## Abstract

Laron syndrome (LS), or primary growth hormone resistance, is a prototypical congenital insulin-like growth factor 1 (IGF1) deficiency. The recent epidemiological finding that LS patients do not develop cancer is of major scientific and clinical relevance. Epidemiological data suggest that congenital IGF1 deficiency confers protection against the development of malignancies. This ‘experiment of nature’ reflects the critical role of IGF1 in tumor biology. The present review article provides an overview of recently conducted genome-wide profiling analyses aimed at identifying mechanisms and signaling pathways that are directly responsible for the link between life-time low IGF1 levels and protection from tumor development. The review underscores the concept that ‘data mining’ an orphan disease might translate into new developments in oncology.

## 1. The Somatotropic Axis and Its Role in Growth Retardation

The critical involvement of the growth hormone releasing hormone (GHRH)-growth hormone (GH)-insulin-like growth factor 1 (IGF1), or somatotropic, axis in normal growth, development, and differentiation has been well established [1,2,3,4]. Likewise, the recognition that aberrations (mostly inherited) in specific components of this endocrine system are correlated with growth pathologies is deeply rooted [5,6]. The GHRH-GH-IGF1 network exhibits an extraordinary level of biological complexity and, not surprisingly, some of the signaling molecules responsible for the growth-promoting actions of the somatotropic axis are also accountable for several key biological processes, including cell division, apoptosis, transcription and translation, etc. [7,8,9].

While many pediatric conditions are correlated with short stature (for a review see Wit et al. 2011) [10], the present review focuses on disorders specifically linked to the GHRH-GH-IGF1 axis. Basic and clinical research conducted over the past half century has identified specific nodes at the hypothalamic, hypophyseal, and other organismal levels whose molecular alterations are directly linked to abnormal growth phenotypes [11,12]. Comprehensive endocrine, biochemical, and genetic analyses of these pathologies has had a huge influence on our understanding of the GHRH-GH-IGF1 system pathophysiology [13,14].

Congenital IGF1 deficiencies are usually defined by low serum IGF1 but normal to high GH levels. These diseases may arise from: (1) GH-releasing hormone receptor (*GHRH-R*) defects [15]; (2) *GH* gene deletion (isolated GH deficiency, IGHD) [16]; (3) GH receptor (*GH-R*) gene deficiency (Laron syndrome) [17]; and (4) *IGF1* gene defects [5,18,19]. Other situations leading to congenital IGF1 deficiency are post-GH-R signaling anomalies (e.g., STAT5 defects), acid labile subunit (*ALS*) mutations [5,20], and the recently described mutation in the PPA2 protein [21]. Table 1 summarizes these molecular defects. On the other hand, disorders associated with IGF1 resistance usually exhibit normal to augmented IGF1 levels. These pathologies may result from mutations of the *IGF1* gene (leading to bioinactive IGF1), IGF-binding protein (IGFBP) abnormalities, mild IGF1 receptor (IGF1R) anomalies, post-IGF1R signaling defects, and end-organ resistance to IGF1 action at the growth plate [22,23,24,25,26].

## 2. Laron Syndrome: A Classical Paradigm of Congenital IGF1 Deficiency

Laron syndrome (LS), also known as primary GH insensitivity, is a type of dwarfism that results from mutation or deletion of the *GH-R* gene. LS may also be caused by post-receptor pathways defects, and it leads to congenital IGF1 deficiency [17,27]. This genetically-transmitted (autosomal recessive inheritance with full penetrance) disease was identified in the mid-1950s in three siblings of Yemenite origin. It was first reported in 1966 [28]. The classical features of LS are: (1) short stature (−4 to −10 SDS below the median normal height); (2) typical face and reduced head circumference; (3) obesity; (4) acromicria (i.e., smallness of the extremities); (5) high basal serum GH; and (6) low to undetectable serum IGF1, without response to exogenous GH [29]. The identification of an exon deletion at the *GH-R* gene as the molecular defect underlying LS etiology was first reported in 1989 [30]. Since this report, several *GH-R* defects have been identified, including exon deletions and nonsense, frameshift, and missense mutations. The majority of the mutations are in the extracellular portion of the receptor, leading to the absence of circulating GH binding protein (GH-BP). Several mutations have been mapped to the cytoplasmic and transmembrane GH-R domains [31,32,33,34]. Despite the variability in the mutations observed, the phenotypic consequences are remarkably similar, i.e., dwarfism, lack of GH signaling, and undetectable, or extremely low, IGF1 values (Figure 1).

In addition to the Israeli cohort in which most initial endocrine and genetic analyses were conducted (comprising now ~75 patients of various ethnic origins), patients with primary GH insensitivity have been reported in Ecuador and in a number of Mediterranean and Middle Eastern countries [35,36]. Of genetic relevance, the same *GH-R* mutation (E180 splice, A to G transition at position 594) was identified in 37 patients from the large Ecuadorian cohort, which is consistent with the notion that this population derived from a single founding ancestor [37,38]. In the Israeli cohort, on the other hand, a number of molecular defects were identified [39].

Treatment of LS patients with recombinant IGF1 (available since the mid-1990s) has been reported to have a significant effect on linear growth acceleration [40]. In addition, growth of the extremities and ’catch-up’ growth of the head circumference have been noticed. However, the growth velocity achieved by IGF1 injections has been found to be less intense than that reached by GH treatment in GH-deficient children [41]. Unfortunately, an initial decline in percent body fat following IGF1 administration has been observed to be followed by increasing adiposity [42,43]. In addition to the obesity associated with therapy, additional side effects have been reported, including tachycardia and skeletal pain, etc.

## 3. Congenital IGF1 Deficiency Confers Protection from Cancer Development

The linkage between high circulating IGF1 dosages and cancer risk has been firmly established by numerous epidemiological studies conducted over the past two decades [44,45,46,47]. This correlation is particularly meaningful in a number of adult epithelial tumors typically linked to endocrine function (e.g., breast and prostate, etc.). In alignment with its strong anti-apoptotic, pro-survival activity, IGF1R is overexpressed in malignantly transformed cells. Increased IGF1R concentrations in tumors is regarded as a critical adaptation that allows *already* transformed cells to rapidly proliferate and progress through the cell cycle. On the other hand, potential correlations between low IGF1 values and cancer incidence have not been investigated in a systematic fashion. A recently-conducted epidemiological study has examined the prevalence of malignancy in a cohort that included 538 congenital IGF1 deficient patients [48,49]. This population was subdivided into: (1) LS patients (*n* = 230); (2) IGHD patients (*n* = 116); (3) patients with *GHRH-R* defects (*n* = 79); and (4) congenital multiple pituitary hormone deficiency (cMPHD) patients (*n* = 113). In addition, analyses included 752 first-degree family members. The study reported that none of the 230 LS patients had a cancer of any type. In addition, only one out of the 116 patients with IGHD had a tumor (Table 2). Eighteen cases of cancer were reported among 218 first-degree family members of LS patients (most of them heterozygotes) (8.3%). Furthermore, twenty-five tumors were reported among 113 further relatives (22.1%). Despite the fact that the number of patients in this cohort was relatively small, differences between the patients and controls were statistically significant. Given that congenital IGF1 deficiencies are rare conditions, the number of patients included in these epidemiological analyses represents a major portion of the entire worldwide population of the diseases.

In a study conducted in Ecuador, Guevara-Aguirre et al. have reported causes of death in LS patients [50]. The cohort used in this study was investigated for more than thirty years and mortality data was collected for 53 LS patients who died before 1988. Tumors were not a main cause of death among LS patients who died before 1988 and there was no proof of cancer among 99 LS patients since 1988. Cancer frequency was similar to the general population among relatives (~20%). Finally, the observations regarding cancer protection in LS were corroborated by animal studies using the GH-R/GH-binding protein (BP) knock-out (KO) (‘Laron’) mouse model [51,52].

The discovery that LS patients are protected from cancer is of major relevance [53]. The interpretation of epidemiological data is in agreement with the concept that the somatotropic axis has a critical role in predisposing progenitor and somatic cells to transformation. IGF1 deficiency, on the other hand, might confer protection against impending development of a tumor. The studies described in this review article were designed to evaluate the hypothesis that life-long lack of exposure to IGF1 in LS activates cancer-protecting pathways, including apoptosis and autophagy. Of importance is the fact that immune deficiency has been reported in association with congenital IGF1 deficiencies [54]. Hence, data suggest that cancer protection in LS is not related to improved immune surveillance but rather to a reduction in the events leading to cancer initiation.

In a broad sense, LS research offers a unique opportunity to address the impact of the GHRH-GH-IGF1 endocrine axis on a number of physio-pathological pathways, including growth, obesity, diabetes, and aging, etc. Obesity constitutes the second major adverse effect of LS (after dwarfism). Obesity starts in utero and continues even during long-term IGF1 treatment [27,29,53]. Body composition analyses showed that fat represents 59% and 39% of body weight in adult females and males, respectively. Body lipids increase with age and this hyperlipidemia often leads to fatty liver. The progressive obesity in this condition correlates, in most cases, with advance from a state of insulin sensitivity in childhood to insulin resistance in young adults and, eventually, Type 2 diabetes mellitus. Finally, while it is difficult to assess the effect of congenital IGF1 deficiency on longevity in a rare condition such as LS, disruption of the GHRH-GH-IGF1 pathway has been shown to be correlated with an extended lifespan in various animal species, including nematode and mouse models.

## 4. Genome-Wide Profiling of Laron Syndrome Patients Identifies Pathways Associated with Cancer Evasion

To discover genes that are differentially represented in LS individuals compared to controls and, in particular, to identify signaling pathways that might be linked to cancer protection for this condition, we recently conducted genome-wide profiling analyses using Epstein-Bar virus (EBV) immortalized lymphoblastoids that were derived from four patients and four controls of the same age range (LS, 44.2 ± 6.1 years; controls, 51.7 ± 11.3 years (mean ± SD)), gender (female) and ethnic origin (Iraq, Iran, and Yemen) [55]. One-way ANOVA was performed using Partek Genomics Suite to create a list of differentially expressed genes. Thirty-nine annotated genes that were differentially expressed in LS compared to controls were identified (*p* value < 0.05; fold-change difference cutoff >|2| (Figure 2a)). Principal component analysis (PCA) revealed very good discrimination between the experimental groups (Figure 2b). Functional analyses provided evidence for a number of pathways that are differentially represented in LS. These enriched signaling pathways include, among other things: cell adhesion, G-protein signaling pathway, cell migration and motility, Jak-STAT signaling, apoptosis, and metabolic pathways, etc. (Table 3 and Figure 3). In general, genes involved in cell cycle control, motility, and growth were down-regulated in LS. As described in the next section, bioinformatics data was validated by biological assays that showed marked differences in proliferation, cell cycle distribution, and autophagy between LS and healthy cells.

Autophagy is an important housekeeping mechanism that is involved in homeostasis maintenance [56,57] by clearing damaged proteins and organelles. Autophagy is also involved in oxidative stress and tumorigenesis. Our analyses revealed that a number of autophagic markers (e.g., LC3β and p62) were differentially expressed in LS cells and suggested that these autophagic adaptations were responsible for the enhanced survival observed in LS cells in response to oxidative stress [55].

## 5. Differential Regulation of Oncogenes and Anti-Oncogenes in Laron Syndrome

Consistent with the epidemiological data described above depicting a markedly diminished cancer prevalence in LS, our bioinformatics analyses demonstrated that lymphoblastoids derived from LS patients express diminished levels of gene transcripts linked to cell cycle progression and oncogenic transformation. These transcripts include, among others: cyclin A1, cyclin D1, serpin B2, versican, and zinc-finger transcription factor Sp1, etc. On the other hand, LS cells express higher levels of tumor suppressors, or anti-oncogenes, that are typically associated with activation of cell protection mechanisms, including pTEN, p21, and others (Figure 4a). Hence, data support the notion that low endocrine IGF1 in LS leads to downregulation of genes that affect proliferation and mitogenesis in a positive fashion, while concomitantly upregulating genes that confer protection from malignant transformation.

In agreement with the well-documented finding that the *IGF1R* gene is overexpressed in most types of cancer [58,59], IGF1R levels (both total and phosphorylated) have been found to be drastically reduced in LS cells [60,61]. This decrease is accompanied by parallel reductions in the phosphorylation of downstream signaling molecules AKT and ERK, two prototypical families of IGF1 and insulin mediators (Figure 4b). We assume that the reduction in expression and activation of components of the IGF1R signaling axis might provide the mechanistic basis for a decrease in the mitogenic potential of LS cells (see below).

Finally, the availability of patient-derived lymphoblastoid cells allowed us to assess the effect of IGF1 deficiency on the in vitro mitogenicity and apoptosis displayed by these cells. Down-regulation of cell cycle genes (e.g., *cyclin D1* and *cyclin A1*) was found to be correlated with altered cell cycle dynamics and apoptosis. Thus, the proliferation rate of LS cells was 50% lower than that of controls. Flow cytometry indicated that the portion of apoptotic cells was 40% higher in LS than in controls (*p* = 0.0005), while the percentage of necrotic cells was augmented by 27% (Figure 4c,d). Combined, results of biological assays provide support to bioinformatics data indicating that LS cells exhibit diminished mitogenic capabilities. The identification of novel molecular targets of IGF1 is described in the following sections.

## 6. Identification of Novel Metabolic Targets for IGF1 Action

Genome-wide profiling of LS-derived lymphoblastoids has revealed an augmented representation of genes involved in defense from toxic xenobiotic substances [62]. These genes include, among others: (1) the uridine diphosphate (UDP) glycosyl transferase gene family (*UGT2B15*, *UGT2B17*; fold-change = 12.4); (2) ribosomal modification protein RimK family member B (*RIMKLB*; fold change = 3.7); and (3) thioredoxin-interacting protein (*TXNIP*; fold-change = 2.35). These genes have not been previously linked to the somatotropic axis.

*UDP-glycosyl transferase gene family*. The UDP-glycosyl transferase gene family (*UDPGT*) plays an important role in the elimination of toxic xenobiotic substances [63,64]. This enzyme exhibits activity towards several classes of xenobiotic substrates, including phenolic compounds, flavonoids, and antraquinones, etc. Our genomic assays demonstrated that the levels of UGT2B15 mRNAs were ~12-fold higher in LS than in control cells. These results were validated by qPCR. The data is consistent with the observation that LS cell survival upon oxidative damage was higher than that of control cells.

*Ribosomal modification protein RimK family member B.* Ribosomal modification protein RimK family member B (RIMKLB) participates in metabolic processes and cellular protein modification [65]. RIMKLB is found in the cytoplasm, where it displays catalytic and ligase activity. RIMKLB also displays metal ion binding activity. Overexpression of RIMKLB in LS might be correlated with catalytic processes as well as apoptotic and autophagic mechanisms.

When combined, the results imply that increased *UGT2B15/UGT2B17* and RIMKLB, among other highly represented metabolic genes in LS, might confer upon these cells: (1) protection against oxidative and genotoxic damage; and (2) more efficient autophagic and apoptotic processes. If substantiated by biological studies, these findings may generate insight into the mechanistic foundation for low cancer prevalence in LS.

## 7. Identification of Thioredoxin-Interacting Protein (TXNIP) as a New Target of IGF1

Thioredoxin-interacting protein (TXNIP) was discovered as a vitamin D3-stimulated gene in leukemia [66]. TXNIP binds to the catalytic site of thioredoxin (TRX) and inhibits its expression. These early results demonstrated the important role of TXNIP in redox regulation [67]. TXNIP also exerts TXR-independent functions, such as regulation of metabolism and cell growth [68]. TXNIP is a member of the α-arrestin family and acts as a tumor suppressor. TXNIP is frequently silenced by genetic or epigenetic mechanisms in cancer cells [69]. Furthermore, TXNIP has a key role in the control of glucose utilization and energy expenditure [70,71]. TXNIP deficiency is also associated with cellular senescence in mice [72]. As mentioned above, TXNIP mRNA levels have been found to be more than 2-fold higher in LS than in healthy cells.

We have recently confirmed the role of TXNIP as a new target for IGF1 and insulin action [73]. Specifically, we showed that IGF1 and insulin inhibits TXNIP expression in several cell lines. Animal studies using GHRKO (‘Laron’) mice confirmed the in vitro experiments. In addition, promoter assays indicated that the effect of IGF1 on *TXNIP* gene expression is mediated at the transcriptional level. Of relevance, oxidative and glucose stresses have been observed to lead to increases in *TXNIP* expression while supplementation of IGF1 has been shown to attenuate TXNIP expression. These results demonstrate that a potential path by which IGF1 exerts its potent antiapoptotic effect is the inhibition of *TXNIP* expression. In view of its tumor suppressor role, we postulate that enhanced *TXNIP* expression in LS might be responsible for tumor protection in this condition. A schematic diagram of the interplay between IGF1 and TXNIP, and the potential implications of this regulatory loop in terms of cell proliferation and homeostasis, is presented in Figure 5. As described below, TXNIP might be relevant in clinics as a diagnostic or predictive biomarker for IGF1R-directed therapies.

## 8. IGFBPs are Differentially Expressed in Laron Syndrome

The role of IGF-binding proteins (IGFBPs) in the regulation of IGF1/IGF2 actions has been extensively investigated [74,75]. IGFBP-1–6 differ in their tissue distribution as well as in their binding affinities for the ligands. In addition, certain IGFBPs have been shown to display IGF-independent actions [76]. The role of IGFBPs in cancer, however, is still controversial [77,78,79,80]. To gain further insight into mechanistic aspects associated with cancer protection in LS, we have assessed the differential representation of IGFBPs in LS-derived lymphoblastoids [81].

Our analyses revealed that IGFBP-2, IGFBP-5, and IGFBP-6 mRNA levels were decreased in LS lymphoblastoids compared to healthy controls by 62%, 75%, and 82%, respectively (Figure 6). IGFBP-4 mRNA levels were similar in patients and controls. On the other hand, IGFBP-3 mRNA levels were increased by 130% in LS cells (*p* < 0.05). Confocal immunofluorescence and Western blots confirmed that differences in mRNA levels were correlated with changes at the protein level.

IGFBP-3 has been portrayed as an anti-oncogene for a number of tumors. We estimate that the increased IGFBP-3 levels in LS are consistent with this role [80]. IGFBP-2 is usually described as pro-tumorigenic, leading to increases in T-cell proliferation [82]. Likewise, IGFBP-5 also promotes T-cell migration while IGFBP-6 functions as a chemotactic agent for T-cells [83,84,85]. Therefore, reductions in these IGFBPs are in agreement with a protective activity against cancer.

## 9. Implications in Personalized Medicine

IGF1R is a promising therapeutic target in oncology [86,87]. Sadly, disappointing results have been obtained when drug candidates, either as monotherapy or in combination with other reagents, were evaluated in Phase III clinical trials. Hence, it is necessary to find biomarkers that can help identify patients who may benefit from IGF1R-directed therapies [88]. In a recent preclinical study, we showed that the mutational status of breast cancer gene 1 (*BRCA1*) may serve as a biomarker for this novel approach. Thus, we demonstrated that: (1) the effect of an IGF1R blocking antibody on inhibition of IGF1-mediated proliferation is reduced in breast cancer cells expressing a mutant *BRCA1* compared to cells expressing a wild-type *BRCA1*; and (2) the synergistic effect of anti-IGF1R therapy along with chemotherapy is similarly reduced in cells containing a mutant *BRCA1* gene [89]. In view of our results showing previously unrecognized links between the IGF1 axis and a series of metabolic (and other) genes that are differentially represented in a rare condition associated with cancer protection, we propose that at least some of these genes may constitute novel biomarkers capable of predicting and/or monitoring responses to anti-IGF1R therapy [90].

In the specific case of the *TXNIP* gene, our data suggest that high TXNIP levels in LS may account for cancer protection in this disease by maintaining cellular homeostasis. Dissection of the complex regulatory loops involving the IGF1 and TXNIP pathways could be of relevance to our understanding of physio-pathological processes as well as to our ability to personalize cancer protocols.

## 10. Conclusions

Genomic, proteomic, and other sophisticated platforms are having a significant impact on our understanding of basic and clinical questions in the field of oncology. Genomic profiling conducted on Laron syndrome patients emphasizes the key role of the GHRH-GH-IGF1 axis in cancer biology. Our analyses have identified new targets for IGF1 action, including a series of metabolic enzymes whose dependence on IGF1 has been previously unrecognized. Future studies will address the transcriptional and epigenetic mechanisms responsible for IGF1 regulation of these novel pathways.

## Figures and Tables

**Figure 1 cells-08-00596-f001:**
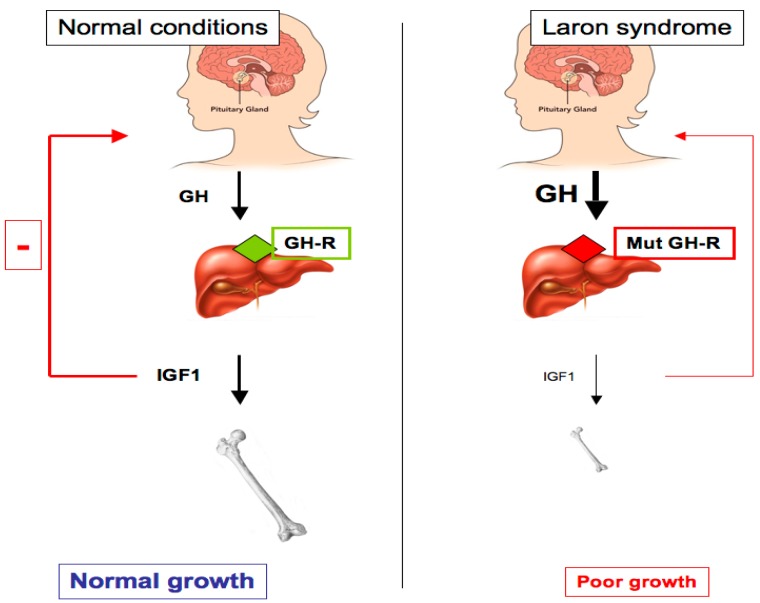
Schematic representation of the GH-IGF1 axis in health and in Laron syndrome (LS) patients. Pituitary-produced GH leads to IGF1 secretion from the liver, with ensuing bone elongation and longitudinal growth (left panel). As a result of a *GH-R* mutation in LS patients, the liver (and, probably, additional extrahepatic tissues) is no longer able to produce physiological levels of IGF1 (right panel). Abrogation of IGF1 production leads to impaired growth and defective negative feed-back at the pituitary gland level, leading to high circulating GH levels.

**Figure 2 cells-08-00596-f002:**
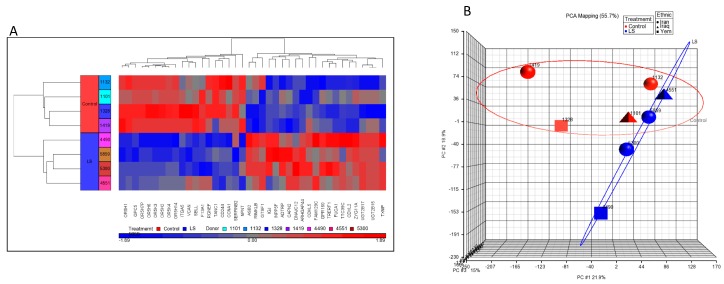
Genome-wide profiling of LS patients. (**a**) Cluster analysis of differentially expressed genes in Epstein-Bar virus (EBV)-immortalized lymphoblastoids derived from four LS patients (four bottom rows, blue color) and four age-, gender-, and ethnicity-matched controls (four upper rows, red color). The figure depicts a cluster of 39 differentially expressed genes (fold change (FC) > 2 or < −2 and *p* value < 0.05). The names of the genes are presented in the *x*-axis. Up-regulated genes are shown in red and down-regulated genes are shown in blue. (**b**) Principal component analysis (PCA) display of four LS and four control arrays. Hierarchical cluster analysis was performed using Partek Genomics Suite software with Pearson’s dissimilarity correlation and average linkage methods. Data analysis was followed by one-way ANOVA. Blue circles: LS patients; red circles: controls. The figure was adapted from [55].

**Figure 3 cells-08-00596-f003:**
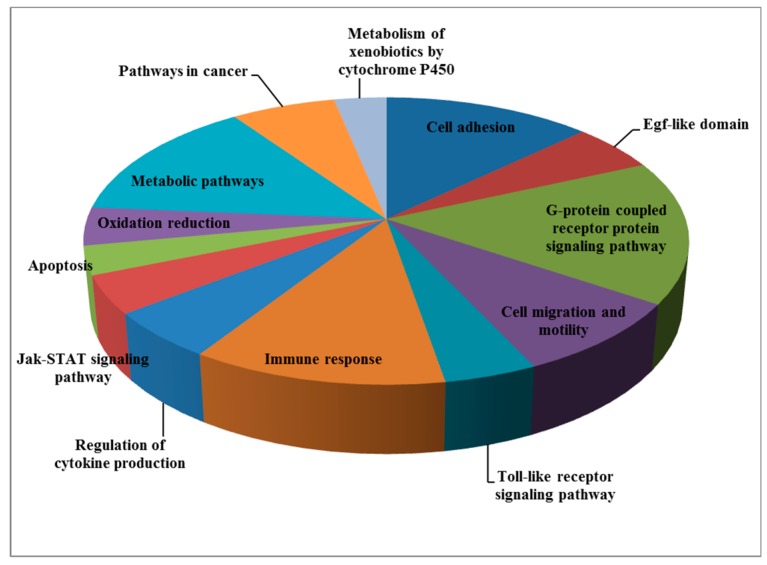
Signaling pathways altered in LS. The pie chart illustrates the signaling pathways that were differentially represented in LS cells as a percentage of the total number of differentially expressed genes.

**Figure 4 cells-08-00596-f004:**
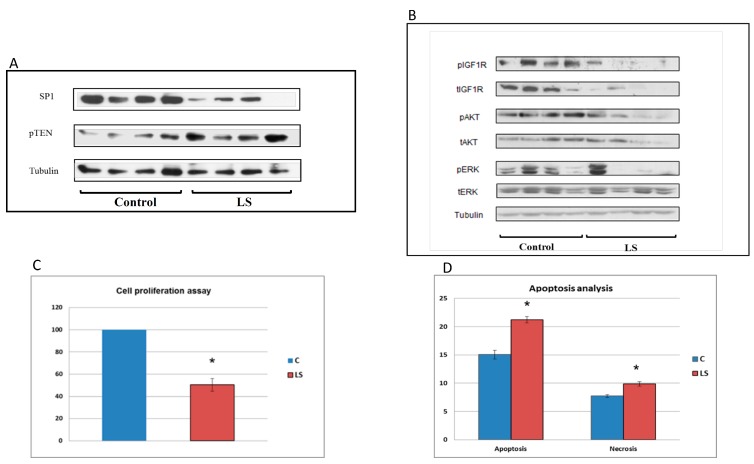
Analysis of signaling pathways associated with cancer protection in LS. (**A**) Western blot analysis of Sp1 and pTEN levels in LS-derived and control lymphoblastoids. Lymphoblastoid cell lines of four LS patients and four controls were lysed and extracts were electrophoresed through SDS-PAGE. Blots were incubated with antibodies against Sp1 and pTEN. The lanes correspond to individual controls and patients. (**B**) Western blot analysis of downstream mediators of IGF1 action in LS. Cell extracts were resolved on SDS-PAGE and membranes were incubated with antibodies against phospho- and total-IGF1 receptor (IGF1R), phospho- and total-AKT and phospho- and total-ERK. Tubulin levels were measured as a loading control. (**C**) Cell proliferation of LS and control cells. Proliferation of LS- and control-derived lymphoblastoid cells was assessed using an XTT colorimetric kit. The statistical significance of differences between groups was assessed by Student’s *t*-test. Legend: *, significantly different versus control (*p* < 0.05); red bars, LS; blue bars, controls. (**D**) Basal apoptosis and necrosis of LS and control cells. Apoptosis and necrosis were measured by flow cytometry analysis after staining cells with an annexin-V antibody and propidium iodide (PI). Necrotic cells were stained with PI as well as annexin V; apoptotic cells were stained only with annexin V. The figure was adapted from [55].

**Figure 5 cells-08-00596-f005:**
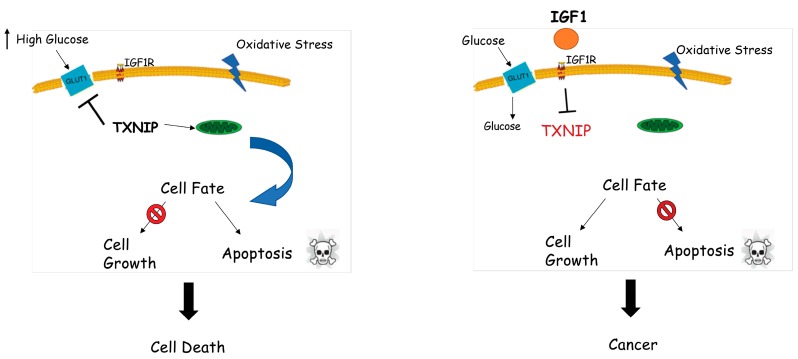
Regulation of thioredoxin-interacting protein (TXNIP) expression by IGF1. The processes of cell survival and homeostasis are tightly controlled by IGF1 action from early ontogenetic stages throughout adulthood. Left panel: normal physiological stress conditions, including starvation and oxidative and glucose stress, might lead to upregulation of TXNIP. Augmented TXNIP levels initiate apoptosis by interacting with thioredoxin and translocating to mitochondria. Cellular stress in the absence of IGF1 (e.g., Laron syndrome) may lead to cell death. Right panel: IGF1 significantly downregulates oxidative and glucose stress-induced TXNIP upregulation and controls glucose uptake in order to improve the energy balance of the cell. Cellular stress in the presence of IGF1 might lead to deregulated cell growth, including cancer.

**Figure 6 cells-08-00596-f006:**
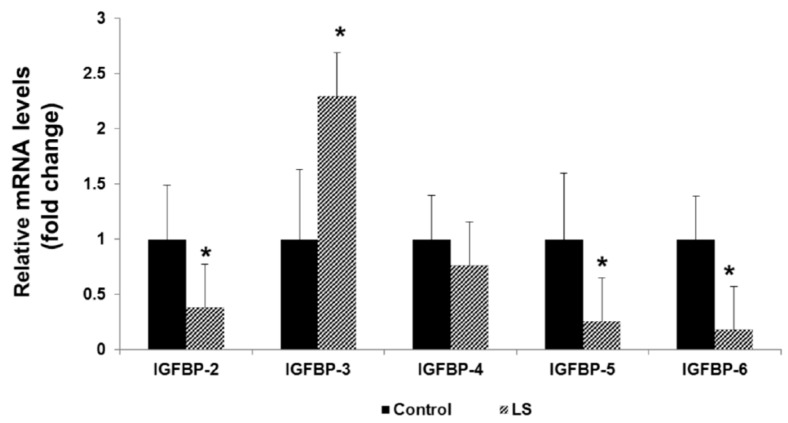
Expression of IGF-binding protein (IGFBP) mRNA in Laron syndrome. Total RNA was prepared from lymphoblastoid cell lines derived from four LS patients (gray bars) and four controls (closed bars) of the same age range, gender, and ethnic origin. Levels of IGFBP-2, -3, -4, -5, and -6 mRNAs were measured by RQ-PCR. For each IGFBP mRNA, a value of 1 was given to the level displayed by controls. Bars denote mean ± SD (*n* = 4). Legend: *, *p* < 0.05 versus respective control. Results indicate that mRNA levels of IGFBPs usually regarded as pro-mitogenic (IGFBP-2, -5, and -6) were reduced in LS, whereas IGFBP-3 (a pro-apoptotic protein) levels were increased under this condition. The figure was adapted from [71].

**Table 1 cells-08-00596-t001:** Molecular pathology of congenital insulin-like growth factor 1 (IGF1) deficiencies.

*Molecular defects leading to congenital IGF1 deficiency*
Growth hormone (GH)-releasing hormone receptor (*GHRH-R*) defect
*GH* gene deletion (isolated GH deficiency, IGHD)
GH receptor (*GH-R*) gene deficiency (Laron syndrome)
*IGF1* gene deletion
Defects of post-GH-R signaling (e.g., STAT5 defects)
Acid labile subunit (*ALS*) mutations
PPA2 protein mutations

**Table 2 cells-08-00596-t002:** Epidemiological analysis of cancer prevalence in LS patients.

	Laron Syndrome	First-Degree Relatives	Further Relatives
Total number (*n*)	230	218	113
Number of malignancies	0	18	25
Prevalence of malignancy	0.0%	8.3%	22.1%

Adapted from Steuerman et al. [48].

**Table 3 cells-08-00596-t003:** Functional analysis of differentially expressed gene clusters in Laron syndrome. The table lists thirteen biological functions that were identified using the David and WebGestalt analysis platform.

Pathway/Function	Number of Genes
Cell adhesion	12
Egf-like domain	5
G-protein coupled receptor protein signaling pathway	15
Cell migration and motility	8
Toll-like receptor signaling pathway	4
Immune response	11
Regulation of cytokine production	5
Jak-STAT signaling pathway	4
Apoptosis	3
Oxidation reduction	4
Metabolic pathways	13
Pathways in cancer	6
Metabolism of xenobiotics by cytochrome P450	3

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
