# Peer review of "Genome-Wide Profiling of Laron Syndrome Patients Identifies Novel Cancer Protection Pathways"

_cells, 2019, doi:10.3390/cells8060596_

Round 1

Reviewer 1 Report

The paper is comprehensive and well written, however the figures need to be adjusted in order for a reader to be able to understand them.

For example; The labels in figure 2 are incomprehensible to the reader, nor are the axis explained adequately in the legend. Further, comments on interpretation of the data, e.g. "PCA revealed a good discrimination between both experimental groups" should not be included in the legend.

Likewise, text/labelling in figure 3 is not clear, and the image quality should be improved so that the data can be interpreted by the reader. The legend could also include more information - for example, what do the 4 lanes per groups illustrate? 4 different patients/controls? 4 different replicates? Also, what statistical test was used to assess significance?

Author Response

1. The legend of Figure 2 was now significantly expanded in order to clarify the information presented in the figure. The sentence indicated by Reviewer 1 was removed from the legend.

2. The information requested by Reviewer 1, including the meaning of the different lanes and the statistical analyses employed, was incorporated into the legend of Figure 3.

Reviewer 2 Report

This article is a review regarding Laron syndrome including its current genome-wide understanding of low cancer prevalence.

Recent updates of the theoretical association between IGF1, IGF1R, and molecular background of carcinogenesis is well documented. However, a theoretical background of so-called cancer evasion in Laron syndrome is still under investigation and unclear so far. Moreover, a major axis of disease reduction of Laron syndrome includes diabetes and a strong association between diabetes and cancer is well known. The prevalence of the combined disease is also an important point of view in carcinogenesis.  

So I strongly recommend performing a more extensive survey for IGF1, IGF-1R, and cancer in diabetes, and their possible relation in carcinogenesis, or even therapeutic implications relation to tumor mutations, and introduce a more theoretical background of low disease prevalence of other diseases like diabetes in this syndrome and its implications on low cancer prevalence, which would reinforce the subject of this paper.

Author Response

We thank Reviewer 2 for this comment. A new paragraph dealing with these important topics was now included in Section 3.

Reviewer 3 Report

This is a good summary of the information gained from studies indicating that subjects with Laron's syndrome have a lower risk of developing cancer. The review is well written and nicely brings together the evidence and what it may tell us about the development of cancer.

In the abstract they describe a recent epidemiological report as "exceptional", this is a rather subjective term and should be moderated to a more objective term such as 'important'.

Author Response

The requested term was replaced in the Abstract.

Round 2

Reviewer 2 Report

The author should add references to paragraph line 137-146.

Author Response

References were incorporated into this paragraph